# Tailoring of Novel Azithromycin-Loaded Zinc Oxide Nanoparticles for Wound Healing

**DOI:** 10.3390/pharmaceutics14010111

**Published:** 2022-01-05

**Authors:** Mohammed S. Saddik, Mahmoud M. A. Elsayed, Mohamed A. El-Mokhtar, Haitham Sedky, Jelan A. Abdel-Aleem, Ahmed M. Abu-Dief, Mostafa F. Al-Hakkani, Hazem L. Hussein, Samah A. Al-Shelkamy, Fatma Y. Meligy, Ali Khames, Heba A. Abou-Taleb

**Affiliations:** 1Department of Pharmaceutics and Clinical Pharmacy, Faculty of Pharmacy, Sohag University, P.O. Box 82524, Sohag 82524, Egypt; mohammed.sherif@pharm.sohag.edu.eg; 2Department of Medical Microbiology and Immunology, Faculty of Medicine, Assiut University, Assiut 71515, Egypt; elmokhtarma@aun.edu.eg; 3Department of Microbiology and Immunology, Faculty of Pharmacy, Assiut University, Assiut 71515, Egypt; haitham.mohamed@aun.edu.eg; 4Department of Industrial Pharmacy, Faculty of Pharmacy, Assiut University, Assiut 71516, Egypt; Jelan.abdelrazik@pharm.aun.edu.eg; 5Chemistry Department, College of Science, Taibah University, Madinah 42353, Saudi Arabia; amamohammed@taibahu.edu.sa; 6Chemistry Department, Faculty of Science, Sohag University, Sohag 82524, Egypt; 7Department of Chemistry, Faculty of Science, Al-Azhar University, Assiut Branch, Assiut 71524, Egypt; dr.mostafa.farouk.83@gmail.com; 8Department of Chemistry, Faculty of Science, New Valley University, El-Kharja 72511, Egypt; 9Department of Dermatology, Venereology and Andrology, Faculty of Medicine, Al-Azhar University, Assiut Branch, Assiut 71524, Egypt; Lotfyh774@gmail.com; 10Department of Physics, Faculty of Science, New Valley University, El-Kharja 72511, Egypt; Samah.ahmed@scinv.au.edu.eg; 11Department Histology, Faculty of Medicine, Assiut University, Assiut 71524, Egypt; Fmeligy@aun.edu.eg; 12Department of Pharmacology and Toxicology, Faculty of Pharmacy, Sohag University, Sohag 82524, Egypt; ali_khamies@yahoo.com; 13Department of Pharmaceutics and Industrial Pharmacy, Faculty of Pharmacy, Merit University (MUE), Sohag 82755, Egypt; Heba.ahmed@merit.edu.eg

**Keywords:** zinc oxide nanoparticles, azithromycin, wound healing, metal nanoparticles

## Abstract

Skin is the largest mechanical barrier against invading pathogens. Following skin injury, the healing process immediately starts to regenerate the damaged tissues and to avoid complications that usually include colonization by pathogenic bacteria, leading to fever and sepsis, which further impairs and complicates the healing process. So, there is an urgent need to develop a novel pharmaceutical material that promotes the healing of infected wounds. The present work aimed to prepare and evaluate the efficacy of novel azithromycin-loaded zinc oxide nanoparticles (AZM-ZnONPs) in the treatment of infected wounds. The Box–Behnken design and response surface methodology were used to evaluate loading efficiency and release characteristics of the prepared NPs. The minimum inhibitory concentration (MIC) of the formulations was determined against Staphylococcus aureus and Escherichia coli. Moreover, the anti-bacterial and wound-healing activities of the AZM-loaded ZnONPs impregnated into hydroxyl propyl methylcellulose (HPMC) gel were evaluated in an excisional wound model in rats. The prepared ZnONPs were loaded with AZM by adsorption. The prepared ZnONPs were fully characterized by XRD, EDAX, SEM, TEM, and FT-IR analysis. Particle size distribution for the prepared ZnO and AZM-ZnONPs were determined and found to be 34 and 39 nm, respectively. The mechanism by which AZM adsorbed on the surface of ZnONPs was the best fit by the Freundlich model with a maximum load capacity of 160.4 mg/g. Anti-microbial studies showed that AZM-ZnONPs were more effective than other controls. Using an experimental infection model in rats, AZM-ZnONPs impregnated into HPMC gel enhanced bacterial clearance and epidermal regeneration, and stimulated tissue formation. In conclusion, AZM -loaded ZnONPs are a promising platform for effective and rapid healing of infected wounds.

## 1. Introduction

Wound healing is a critical process that involves four stages, including hemostasis, inflammation, proliferation, and maturation [1,2]. Any interruption of these stages will retard the repair of damaged tissues. The healing process can also become impaired and complicated because of infection by pathogenic bacteria that colonize the damaged tissues, with increasing incidence of spreading the infection and systemically leading to fever and sepsis [3]. So, there is an urgent need for the development and discovery of novel materials that cure infected wounds and accelerate the healing process.

Nanomaterials have a great intrinsic potential for preventing and treating wound infections, with multiple advantages compared with traditional treatment approaches [3,4]. Metal and metal oxide nanoparticles [5,6,7] exhibit remarkable antibacterial activities against different pathogens, in addition to their low toxicity profile [8,9].

ZnO is a bio-safe material that possesses multiple functions, including photo-oxidizing and photocatalysis impacts on chemical and biological species [10,11]. Additionally, its UV-blocking activity has directed its use in sunscreen preparations [12]. Many reports have shown the antibacterial activity of zinc oxide nanoparticles (ZnONPs) against both Gram-negative and Gram-positive bacteria. Furthermore, ZnONPs are considered ideal candidates for carrying pharmaceutically active molecules [13].

Several approaches can be used for the synthesis of ZnONPs as physical fragmentation and chemical, such as sol-gel, vapor condensation, amorphous crystallization, green methods, and many others [14,15,16,17,18].

Alternatively, AZM belongs to the macrolide group of antibiotics. This group of antibiotics selectively binds to the bacterial 50S ribosomal subunit, thereby inhibiting the vital process of protein synthesis. It is a dicationic molecule that is characterized by its ability to permeate the bacterial outer membrane of bacteria, which is an important step in overcoming bacterial self-defense [19].

Many studies have reported remarkable benefits of the adsorption of drugs on the surface of metals and metal oxide nanomaterials, such as adsorption of phenytoin on copper nanoparticles for wound healing [20] and 5-Fu on chromium nanoparticles for colorectal cancer treatment [21]. This is the first study to report on the adsorption of AZM onto the surface of ZnONPs. Considering the interesting findings of ZnONPs and AZM in the healing process of infected wounds, the major objective of our study is to explore the effect of AZM-ZnONPs for treating infected wounds. AZM-ZnONPs were characterized for morphology, FT-IR, and XRD. Moreover, AZM adsorption isotherm onto a ZnONP surface was investigated. The in vitro antibacterial activities were evaluated. Additionally, the in vivo antibacterial and wound-healing activities of AZM-loaded ZnONP-impregnated HPMC gel was also performed using an excisional wound model in rats.

## 2. Material and Methods

### 2.1. Employed Reagents in the Current Investigation

All the chemicals used in the current research were analytical grade and were not further purified. HPMC was used as a gelling agent and purchased from Sigma-Aldrich (St Louis, MO, USA). AZM (AZM) was used as the adsorbate and was received as a gift from El-Nile Company for Pharmaceutical Industries (Cairo, Egypt). Zn(NO_3_)_2_ 6H_2_O (Sigma-Aldrich) was the zinc precursor, whereas sodium bicarbonate NaHCO_3_ (98%, Alfa Aesar) was the precipitating agent, and solvents (de-ionized water) were used for washing the formed precipitate.

### 2.2. Methods

#### 2.2.1. Synthesis of ZnONPs

ZnONPs were prepared by a combination of precipitation and calcination methods. One gram of zinc nitrate Zn(NO_3_)_2_ 6H_2_O was dissolved in 30 mL bi-distilled water and magnetically stirred until the reagents were fully dissolved. After that, a drop-by-drop solution of NaHCO_3_ (2 M) was added to the dissolved mixture with continuous stirring until pH = 7 was reached. The resulting product was further agitated for 1 h, centrifuged, and washed several times with de-ionized water before being dried at 120 °C for 12 h. The powder was ground to pass through meshes before being placed in a crucible for calcination at 800 °C for 2 h at a rate of 10 °C/min. Finally, the ZnONP powder obtained was used for further analysis (Figure 1).

#### 2.2.2. Adsorption of AZM by ZnONPs

In this study, the drug concentration effect on the adsorption performance was evaluated using the batch-adsorption mode. A stock solution of 1 mg/mL in absolute alcohol as a solvent of AZM was prepared. Then, serial dilutions of AZM concentration at 0.1, 0.2, 0.3, 0.4, and 0.5 mg/mL were conducted from the stock solution to determine the drug concentration that indicated the highest removal percentage. The experiments were performed in a 100 mL Erlenmeyer conical flask with a final solution volume of 40 mL with fixed stirring at 500 rpm at room temperature (25 ± 1 °C) using 100 mg ZnONPs for 5 h. Subsequently, the solution was centrifuged at 10,000 rpm and filtered via the vacuum pump using a nylon filter paper of 0.45 µm. The unbound AZM concentration was measured by the previously reported HPLC method [22]. The adsorption performance was repeated for three replicates for all experiments (as standard without ZnONPs and tested with ZnONPs) to assess the robustness of the method repeatability according to quality control and method validation guidelines [23,24,25,26,27]. The results were obtained as the mean of three experimental replicates. The drug removal *P* (%) and the adsorbed loaded amount “adsorption capacity” at equilibrium conditions, *q*_e_ (mg/g), were calculated as the equations in Table 1.

The Langmuir, Freundlich, Temkin, and D-R isotherm models were also conducted on the experimental results to determine the appropriate isothermal model as the following equations.

#### 2.2.3. X-ray Diffraction, Scanning Electron Microscope, and Fourier Transform-Infrared (FT-IR) Analysis

The crystallographic structure and phase composition of pure ZnONPs was investigated by X-ray diffraction (XRD) (Philips X’pert MRD diffractometer, Westhorst WH121, The Netherlands). At room temperature (RT), the diffraction peaks were detected using Cu-Ka radiation (1.5418) with glancing incidence and an angle of incidence of 0.75°. All data were gathered at 2 from 10° to 80° with a step of 0.05° and a counting time of 2.5 s/step for typical powder patterns.

The chemical composition of the samples was examined using a scanning electron microscope (SEM) (model: JEOL-6610VL, Tokyo, Japan) fitted with an X-Max Silicon Drift Detector for energy dispersive X-ray spectroscopy (EDXS, Oxford, UK) analysis and a 20 kV accelerating voltage [32,33]. The size of the nanoparticles in the samples was measured using transmission electron microscopy (TEM) (model: JEOL-JEM-2100F Tokyo, Japan) with a field emission gun and a 200 kV accelerating voltage (FEG). The Fourier transform-infrared (FT-IR, Perkin-Elmer, Norwalk, CT, USA) analysis of ZnO and AZM-ZnONPs was collected at RT using the KBr pellet disc technique in the wavelength range of 400–4000 cm^−1^.

#### 2.2.4. In Vitro Drug-Release Study

In vitro release of AZM from the formulated ZnONPs was performed using the dialysis release method as previously reported [34,35,36] with some modification. Two mL of AZM-loaded ZnONP suspension containing 0.2 mg of AZM was placed on a dialysis membrane (MW cut off 12,000 Da) firmly stretched over one end of a glass cylindrical tube (3 cm^2^ diameter). The tube was then immersed in a 100 mL beaker containing 50 mL phosphate buffer saline (PBS) at pH 7.4 and placed in a thermostatic water bath at 37 °C and shaken at 100 rpm. At predetermined time intervals, 5 mL of sample were withdrawn and replaced by an equal volume of PBS to maintain the sink condition. The amount of AZM released was determined by HPLC as previously reported.

#### 2.2.5. Experimental Design

The formulation characteristics of AZM-ZnONPs were examined and optimized using a Box–Behnken experimental design (BBD) for the maximum LE percentage and rapid drug delivery after 1 and 3 h [21,37,38,39,40,41]. In circumstances involving more than two dependent variables, this design was chosen since it necessitates fewer treatment combinations than other designs [42,43,44]. The BBD is also ratable and has statistical “missing corners,” which can help the experimenter avoid combination factor excesses. In such instances, this characteristic prevents data loss [45]. AZM-ZnONPs were prepared using a three-factor, three-level design. AZM concentration (X_1_), ZnONP weight (X_2_), and temperature were the three independent formulation factors investigated during the study (X_3_). Table 1 shows the variables that were chosen, with their real and coded values according to the plan. AZM-ZnONP formulae were developed based on this design (D1–D15). In the preceding design, three levels of AZM concentration were used: 0.25, 0.5, and 0.75%, signified by the numbers 1, 0, and +1, respectively. The weights of ZnONPs applied were 300, 400, and 500 mg, which corresponded to the values −1, 0, and +1, respectively. Eventually, the reflux temperature was set to 20 ± 1 °C, 25 ± 1 °C, and 30 ± 1 °C, expressed by the numbers −1, 0, and +1, respectively. The loading efficiency % (Y_1_), release in 1 h (Y_2_), and release in 3 h (Y_3_) were chosen as the dependent variables to be examined for the produced AZM-ZnONPs.

#### 2.2.6. Preparation of AZM, Plain ZnONPs, and AZM-Loaded ZnONP-Impregnated HPMC Gels

HPMC was selected as the gelling agent at a concentration level of 2.5% (*w*/*v*) in accordance with a previously reported procedure [20]. Briefly, HPMC gel was prepared by dispersing the polymer in preheated distilled water (80 °C). Then, the prepared dispersion was stirred at 1000 rpm using a magnetic stirrer (Thermo Fisher Scientific, NJ, USA) until a homogeneous mixture was attained. To the prepared gel base, accurate amounts of AZM (0.01% *w*/*w*), AZM-loaded ZnONPs (0.0623% *w*/*w*), and plain ZnONPs (0.0623% *w*/*w*) were added and stirring was continued until a homogenous distribution was obtained. Finally, the prepared gels were kept in the refrigerator at 4 °C until the next use.

#### 2.2.7. In Vitro Antibacterial Activity

The in vitro antibacterial activity of the AZM-ZnONPs was initially tested using the agar diffusion method against clinical isolates of methicillin-resistant *Staphylococcus aureus* (MRSA) and *Escherichia coli* (*E. coli*). A freshly prepared bacterial suspension of 0.5 McFarland turbidity was inoculated on the surface of Mueller–Hinton agar plates and 1 cm wells were prepared using a cork borer. Agar was removed from the wells and AZM, ZnO, AZM-ZnONPs, and saline solution (negative control) were added to the formed wells. Plates were incubated at 37 °C overnight and the diameters of the inhibition zones were recorded [46,47].

#### 2.2.8. Determination of the Minimal Inhibitory Concentration (MIC)

The MICs of the AZM, ZnO, and AZM-loaded ZnONPs were determined against the clinical MRSA and *E. coli* isolates using the broth dilution technique [48]. Firstly, different formulations were serially diluted in a 96-well plate in Mueller–Hinton broth starting at a concentration of 200 µg/mL. Then, the bacterial suspension was added to each well, the plate was incubated at 37 °C for 24 h, and the MICs were determined based on the determination of the lowest concentration showing visualized growth inhibition.

#### 2.2.9. In Vivo Antibacterial and Wound-Healing Efficacy

The in vivo antimicrobial and wound-healing activities of AZM and AZM-ZnONPs were tested in an excision wound model in rats [49]. Rats were housed at room temperature (25 ± 1 °C) and subjected to light/dark cycles. Experiments were conducted in accordance with the internationally accepted Guidelines for the Care and Use of Laboratory Animals [50] and the protocol was approved by the Faculty of Pharmacy’s Ethics Committee (ES06/2020). Briefly, 20 albino rats were divided into four groups, each containing five rats. After intraperitoneal thiopental anesthesia, the skin on the dorsal surfaces was shaved and a 1 cm^2^ circular incision was carefully made using sharp sterile scissors. Five minutes later, the wound was inoculated with 100 µL of MRSA suspension containing approximately 10^8^ colony-forming units (CFU). After 24 h of wound inoculation, groups I, II, and III received treatment two times daily with AZM, ZnO, and AZM-ZnONPs impregnated in 2.5% HPMC gel preparations, respectively, whereas the wounds of non-treated control group IV received blank non-medicated HPMC gel (untreated control group). On days 1, 3, and 7 after wounding, the wounds were photographed and a swab was obtained to determine the bacterial load. The bacterial swabs were spread on the surface of mannitol salt agar and incubated at 37 °C for 24 h to determine the number of CFUs. In addition, for the duration of 10 days after infection, the wound area and wound contraction were determined daily to evaluate the effect of the treatment on the wound-healing process, as previously described [49,51].

#### 2.2.10. Histological Evaluation

Skin specimens were fixed in 10% formalin, dehydrated, embedded in paraffin, and then were cut into 5 μm sections. Staining with hematoxylin and eosin (H&E) stain was done to determine the general skin structure and histological changes in the epidermis and dermis. A group of uninjured rats (negative control group) was included in the study to elucidate the normal skin characteristics and for comparison purposes.

Masson’s trichrome dye was used to see the collagen fibers. Using computerized image analyzer system software (Leica Q 500 MCO; Leica, Wetzlar, Germany) attached to a camera with a Leica universal microscope, the thickness of the epidermis was calculated on H&E-stained sections. The measurements were done using a ×40 objective lens. Each group had five serial sections taken. Measurements were taken in six nonoverlapping fields in each region. The epidermal thicknesses were measured in µm using the arbitrary distance technique, which involves drawing a straight line from the foundation membrane to the epidermis’ top limit [52,53]. The average of six separate area measurements was calculated. Additionally, the mean percent of the area of collagen fibers in the dermis was determined using Masson’s trichrome. The measurements were done in six nonoverlapping fields from five different sections in each group with a ×10 objective lens.

## 3. Results and Discussion

### 3.1. FT-IR Analysis

The chemical bonding and purity of our analyzed materials were further checked using FT-IR analysis. Figure 1A and Table 2 depict the raw FTIR-transmittance spectra of pure ZnO, AZM, and AZM-ZnONPs in the range of 400–4000 cm^−1^ at RT. The characteristic Zn–O stretching mode for wurtzite ZnO was identified in the investigated ZnO and AZM-ZnONPs in the far infrared region at wave numbers between 430 and 445 cm^−1^. These stretching modes below 500 cm^−1^ are suggestive of the successful production of ZnONPs [54]. The O-C-O bending mode of carbonate was ascribed to the band centered at 877–880 cm^−1^, which is commonly found when FT-IR samples are examined without vacuum [55]. The O-H stretching and O-H-O bending vibration modes of surface-adsorbed molecules of water were ascribed to the substantial transmittance bands at 3415–3415 cm^−1^ and 1625–1664 cm^−1^, respectively [56].

Various AZM characteristic bands could be identified in the spectra acquired straight from the solid standard, as shown in Figure 1A. The main bands identified were (i) bands associated with the axial stretching and bending of C-H of the methyl groups, which were found in the ranges of 2966–2985 cm^−1^ and 1367–1374 cm^−1^, respectively [57,58]; (ii) the sharp and powerful band at 1710–1722 cm^−1^, which could be attributed to the axial stretching of the lactone’s C=O group; and (iii) other notable bands in the spectra, including those found in the 1130–1225 cm^−1^ region, which emerged due to absorption related to C-O axial stretching [59].

### 3.2. XRD Analysis

XRD analysis was used to analyze the phase crystalline nature and purity of the produced ZnO powder sample. The XRD patterns of ZnONPs are shown in Figure 1B and XRD data in Appendix A. The sample showed strong and sharp diffraction peaks, suggesting that the ZnONPs crystallized to a great degree. The hexagonal (wurtzite) phase of crystalline ZnO, the space group P63mc with lattice constants a = 3.253 and c = 5.213, (Standard JCPDS file No. 76-0704), could be well indexed in the XRD spectra; no other supplementary crystalline phases linked to contaminants were detected by XRD, showing a high wurtzite phase purity of the as-synthesized ZnONPs. Furthermore, the largest peak intensity of the ZnONP sample was at 2*θ* = 36.275°, which was associated with 101 Miller indices associated with the wurtzite ZnO phase.

The average crystallite size (*D_v_*) of the ZnO sample was estimated using the well-known Debye–Scherer formula [60,61,62]:(11)$$D_{v}=\frac{0.94\lambda }{{\left(FWHM\right)}_{}\cos\theta }$$
where λ is the wavelength of the x-radiation source (i.e., λ = 1.5418 Å), FWHM is the full width at half maximum in radians along the 120 plane, and *θ* is the diffracted Bragg angle. The crystallite size determined by XRD analysis (38 nm) matched the usual particle size determined by TEM examination (39 nm).

### 3.3. EDAX Analysis

To investigate the elemental and compositional stoichiometry properties of the investigated ZnO sample, the chemical composition ZnONP sample was analyzed using energy-dispersive X-ray spectroscopy (EDXS). Figure 1C indicates the typical EDAX spectra of pure and mixed ZnO nanostructure samples. It is clear from the observed EDAX spectra that the prepared powder sample was composed of zinc (Zn) and oxygen (O) without any impurities, which confirmed the high purity of the prepared powder sample.

### 3.4. Microstructural Studies by SEM and TEM Analysis

SEM and TEM investigation confirmed that the ZnONP sample had a sphere-like morphology with monodispersion nanoparticles. Typical SEM and TEM micrographs of ZnONPs and AZM-ZnO samples clearly demonstrated the large aggregates of near spherical shapes with different particle sizes of the as-synthesized samples, as shown in Figure 2. It is clear that the size and the morphology of the particles were little affected by the adsorption of AZM on ZnONPs. In addition, Figure 2D,F show the corresponding diameter distribution histogram of the ZnONP and AZM-ZnO samples estimated by Image J software and Excel calculation [63,64,65]. From TEM analysis, the calculated mean particle sizes matched well with the values that were obtained from the XRD analysis.

### 3.5. Adsorption of AZM by ZnONPs

The effect of drug concentration on the removal performance is shown in Figure 3. As the drug concentration increased from 100 to 200 mg/L, the drug removal decreased from 78.0 to 66.2%. The decrease in the removal depended on the drug concentration and empty sites of the adsorbent. At high concentrations of the drug, the drug adsorbed on the empty sites of the adsorbent until a saturation state was achieved [20] (Figure 2). When the drug concentration continued to increase from 300 to 500 mg/L, the removal percentage slightly decreased from 59.0 to 55.9%. Nevertheless, the decrease can be neglected. Moreover, the drug removal percentage depending on the drug concentration showed a constant trend after a drug concentration of 300 mg/L.

#### Adsorption Isotherm Studies

The four different isotherm models were used to determine the most suitable model for design purposes. The isotherm calculations were done as shown in Table 3.

According to the R^2^ values given in Table 3 and Figure 4, the most fitted isotherm model was determined to be the Freundlich isotherm. According to the Freundlich model, the adsorbent was heterogeneous on the surface with a multilayer adsorption capacity. We could also estimate some important parameters of the adsorption study according to the manifested data from different implemented isothermal models, such as the following.

The isotherm adsorption process of AZM on the ZnONP surface could be arranged as follows: Freundlich > Temkin > Langmuir > D-R model using the R^2^ values as an indication parameter.

The Freundlich model indicated the type of adsorption where it was found to be desirable and favored according to the n parameter that was found (1.847) between 1 and 10. The 1/n value was also found to be less than the unit, at 0.541, which revealed that adsorption follows a heterogeneity pattern at the ZnONP surface. According to the Langmuir model, q_L_ (160.4 mg/g) was found to be more than the experimental q_e_ (111.7 mg/g). The adsorption process was also found to be favorable according to the R_L_, where R_L_ was more than 0 and also less than the unit and the actual value was found to be 0.196. As for the Temkin model, the heating of the adsorption process of b_T_ (74.06 kJ/mol) was found to be less than 80 kJ/mol, which means the adsorption was physisorptive in nature [20,28]. The D-R isothermal model was also investigated and gave us an indication about the average of the adsorption energy (E_D_) for each AZM molecule that adsorbed using ZnONPs. It was found to be 77.2. Since it was less than 80, it is indicates that the adsorption could be expressed as a physisorption type [66].

### 3.6. Drug-Loading Efficiency on ZnONPs (LE%)

Table 4 shows the impact of selected variables on LE percentage and drug release after 1 and 3 h. In terms of AZM concentration, the determined theoretical drug content value for D1, D5, and D12 was 20%, whereas it was 33.33% for D2, D3, D7, D8, D9, D13, and 14 and 42.85% for D4, D10, D11, and D15. Using the ANOVA test, the impact of increasing the AZM concentration on the LE percentage was examined for each AZM concentration, ZnONP weight, and temperature. The LE percentage ranged from 50.07 ± 1.93% to 87.11 ± 3.53%, depending on the AZM concentration, ZnONP weight, and temperature. The fitted LE percentage model regression equation is [38,43]:Y_1_ = 74.55 + 7.36 X_1_ + 6.62 X_2_ − 5.14 X_3_ − 1.19 X_1_^2^ − 1.47 X_2_^2^ + 5.22 X_3_^2^ − 2.32 X_1_X_2_ + 1.66 X_1_X_3_ − 0.44 X_2_X_3_(12)

At confidence levels of 90% and higher, there was a statistically significant correlation between the variables since the *p*-value in the ANOVA table was less than 0.1 [44]. According to the R-squared value, the model (as fitted) accounted for 94.21% of the variation in AZM concentrations. The output, on the other hand, provided the results of using a multiple linear regression model to explain the association between ZnONP weight and LE percentage. At the 90% confidence level, there was a statistically significant link between the variables since the *p*-value was less than 0.1. According to the R-squared value, the model (as fitted) explained 83.80% of the variation in ZnONPs weight. Despite the drop in LE% with increasing temperature, the temperature had no significant influence on LE% since the *p*-value was more than 0.1.

With a fixed ZnONP weight and temperature, the LE percentage rose from 50.07 ± 1.93 to 69.72 ± 2.78% and from 69.33 ± 2.39 to 82.34 ± 2.52% for lower and higher AZM concentrations, respectively (Figure 5). Because more of the drug was adsorbed and assimilated on the surface of ZnONPs with high AZM concentrations, LE% improved with higher AZM levels. The % drug incorporation was studied using a linear connection between the two variables. The two key stages in the synthesis of AZM-ZnONPs were the production of stable ZnONPs and the subsequent adsorption of AZM on the surface of the ZnONPs. These two procedures had a significant impact on the size and LE percentage of nanoparticles produced.

With a fixed AZM concentration and temperature, LE% increased from 50.07 ± 1.93 to 69.33 ± 2.39% and from 70.44 ± 2.98 to 81.89 ± 2.81% at lower and higher ZnONP weights, respectively (Table 4). The surface area available for drug loading grew as the weight of the ZnONPs rose. Because of the increased surface area, more drugs could be adsorbed on the nanoparticles’ surface.

The temperature employed during nanoparticle manufacturing affected the LE% of the drug; as the temperature increased, the LE% decreased. Because the drug-adsorption process on the surface of the ZnONPs decreased as the temperature of the AZM with the ZnONP reflux process increased, these results were expected [59]. The major goal of the temperature impact study was to find the optimal reflux temperature that would result in a greater LE percentage. With a steady AZM concentration and ZnONP weight, the LE percentage fell from 70.44 ± 2.98 to 62.31 ± 1.73% (D2, D3) and from 75.25 ± 2.21 to 62.42 ± 3.15% (D5, D6) at lower and higher temperatures (Table 4), respectively. All of these findings suggest that a lower temperature (20 ± 1 °C, −1) should be employed to get larger LE% values.

### 3.7. In Vitro Release Study of AZM-ZnONPs

The outputs of the correlation between AZM concentration, ZnONP weight, and temperature with the percentage drug released after 1 and 3 h is shown in regression Equations (13) and (14), respectively.
Y_2_ = 27.76 + 2.62 X_1_ + 0.70 X_2_ − 2.10 X_3_ − 3.59 X_1_^2^ − 2.31 X_2_^2^ − 0.69 X_3_^2^ − 1.71 X_1_X_2_+ 0.19 X_1_X_3_ − 1.93 X_2_X_3_(13)
Y_3_ = 73.06 + 6.65 X_1_ + 1.76 X_2_ − 5.75 X_3_ − 5.35 X_1_^2^ − 6,43 X_2_^2^- 1.16 X_3_^2^ − 3.14 X_1_X_2_ + 0.32 X_1_X_3_ − 2.60 X_2_X_3_(14)

After 1 and 3 h, the AZM concentration, as well as the ZnONP weight, had a significant influence on drug release (*p*-value = 0.1). The fraction of drugs released rose as their levels rose. The temperature had no influence on drug release, in contrast to its effect on the LE percentage.

The response surface plots of the in vitro release of AZM from its ZnONPs are shown in Table 4 and Figure 6 and Figure 7. The in vitro release of AZM from formulas D1, D2, D3, and D4 employing ZnONPs (X_2_) at a constant level (−1) with varied AZM concentrations (X_1_) was 0.25% for D1, 0.5% for D2 and D3, and 0.75% for D4 (X_2_).

At the completion of one hour, the highest and minimum percentages discharged were found to be 22.42 ± 1.38 and 15.46 ± 1.38%, respectively (Y_2_). After 3 h (Y_3_) of dissolution, the maximum and minimum in vitro releases were 65.44 ± 2.72 and 47.53 ± 1.66%, respectively. In terms of in vitro release within 3 h, the formulae investigated can be ranked as follows: D2 > D4 > D3 > D1 in descending order.

Table 4 shows the summary of in vitro AZM release from ZnONPs using formulae D5, D6, D7, D8, D9, D10, and D11 employing constant ZnONP weight (X_2_) at the medium level (0), variable temperature levels (range from −1 to +1), and various different AZM concentrations (X_1_).

At the end of one hour, the highest and minimum percentages released were 30.89 ± 1.73 and 18.43 ± 1.74% (Y_2_), respectively. After 3 h of dissolution (Y_3_), the maximum and minimum in vitro release were determined to be 77.16 ± 2.96 and 53.54 ± 1.86%, respectively.

The release of AZM from ZnONPs with formulae D12, D13, D14, and D15 in vitro was investigated using constant ZnONP weight (X_1_) at the medium level (0), variable temperature levels (ranging from −1 to +1), and various different AZM concentrations (X_1_).

Equations (13) and (14) show that the weight of ZnONP had a considerable favorable influence on drug release after 1 and 3 h in different formulations (9). At constant (X_1_) and (X_3_) (0, −1), the percentage of AZM released at the end of 3 h (Y_2_) was 65.44 ± 2.71 and 75.23 ± 2.75% (D2, D13) at lower and higher ZnONPs weights, respectively.

All of these findings show that raising the AZM concentration had a substantial impact on drug release. These findings are linked to the ease with which AZM can be released since the drug was adsorbed on the surface of ZnONPs and a larger amount of the drug was accessible for release. The difficulties of desorption of AZM from ZnONPs in the release-medium environment, as well as the nature of the produced AZM-ZnONP combination, might explain these results. AZM desorbs quickly at an alkaline pH, and the amount released increased considerably due to the basic composition of the AZM and ZnONPs, which permitted more of the drug to be accessible for absorption sites. In a study by Zora Rukavina et al. [67], controlled-release AZM liposomes exhibited a sustained drug-release pattern after an initial burst release.

The weight of ZnONPs had an inverse relationship with the percentage of drugs released. These findings are indicative of the fact that when the NP weight increased, the surface area of ZnONPs increased, allowing more medication to be adsorbed on the surface of the monolayer pattern. This AZM monolayer adhesion had a deleterious impact on drug release from the ZnONP surface. The quantity of drug adsorbed in a multilayer pattern on the surface of ZnONPs, which made it more easily released because of attachment forces, decreased whenever the ZnONP weight was lowered [27,68].

Temperature showed no impact on AZM release after 1 and 3 h, in contrast to the LE percentage [42,69]. The findings of the LE percentage on in vitro release after 1 and 3 h suggest that the best AZM-ZnONPs formula should be made with a high AZM concentration (+1, 0.75%), a medium ZnONP weight (0, 500 mg), and a low temperature (−1, 20 ± 1 °C). The optimum formula was D9, according to the rank order of all formulas (considering the greater LE percentage and Rel. 1 and 3 h), and was selected for further in vivo studies.

### 3.8. In Vitro and In Vivo Antibacterial Activities

To evaluate the antibacterial activity of the preparations, the agar diffusion method was performed. The mean inhibition zone diameters were related to the free formulation of AZM.

A concentration of 100 µg/mL of free AZM produced about 13 ± 3 mm and 9.5 ± 2 mm against MRSA and *E. coli*, respectively. However, the same concentration of AZM-ZnONPs produced 20 ± 4 mm and 14 ± 3 mm against MRSA and *E. coli*, respectively, pointing to a higher antibacterial effect. In addition, MIC of the free drug and nano formulation was determined. The lowest inhibitory concentration of AZM was significantly reduced from 20 µg/mL to 10 µg/mL and from 32 µg/mL to 20 µg/mL in the AZM-ZnONPs against both the Gram-positive MRSA and the Gram-negative *E. coli*, respectively (Figure 8).

To evaluate the in vivo anti-microbial and wound-healing characteristics of the formulations, different groups of wounded rats received daily treatment regimens of AZM (Group I), ZnO (Group II), or AZM-ZnONPs (Group III) and were compared with Group IV, which received a non-medicated gel. To visually elucidate the effect of AZM-ZnONP treatment, different groups of treated animals were photographed at 1, 3, and 7 days post infection (Figure 9A). Group III exhibited faster wound closure compared to other groups that received the other treatments, as demonstrated by the higher level of wound contraction percentage. Group III showed better skin appearance and normal hair growth compared to other treated rats or the control group IV, which showed a dramatic progression of the wound area with reduced healing.

Furthermore, bacterial counts were significantly decreased in wounds of rats treated with AZM, ZnO, or AZM-ZnONPs after three and seven days of infection compared to the control group. However, the reduction in bacterial counts in Group III, which received the AZM-ZnONP formulation, was significantly higher than those achieved when rats were treated by other protocols at three and seven days post infection (*p* < 0.01 and *p* < 0.001, respectively). As expected, Group IV did not show a significant improvement in bacterial levels (Figure 9B).

In addition, the gel containing AZM-loaded ZnO nanoparticles (Group III) showed an increased rate of wound contraction compared to the formulation containing free AZM (Group I). On day 10 post infection, AZM-ZnONPs induced a wound healing of 95.8 ± 9.8% compared to 71.5 ± 8.9% in the case of the AZM-treated group or 38.6 ± 8.1% in the blank group (Group IV) that received non-medicated gel (Figure 9C).

Antibacterial activities of zinc oxide nanoparticles and AZM has received significant global interest particularly via the application of nanotechnology to develop particles in the nanometer range. According to our results, the combination of AZM and ZnONPs resulted in improved antimicrobial and wound-healing potencies, which may have resulted from the enhanced physicochemical characteristics of the formulation. Analyzing the MIC of the different formulations showed that the AZM-ZnONP formulation was very effective against both MRSA and *E. coli* and the MIC was reduced markedly compared to each of them when used alone. Similarly, Aljihani, et al. [48] reported that liposomal AZM and liposomal AZM/N-acetyl cysteine markedly improved the MIC of AZM against *E. coli*. Another study showed that AZM nanoparticles produced the same inhibitory effect against *E. coli* at an eight times lower concentration compared to free AZM [70]. In another approach, cefixim and AZM were synthesized by anti-solvent precipitation and the synthesized nanoparticles showed better antibacterial activities against *S. aureus*, *E. coli*, Salmonella typhi, and Shigella as tested by agar diffusion assays than the parental drugs [71]. Furthermore, PLGA-loaded AZM nanoparticles had improved antibacterial potential for treating intracellular infections compared to free drugs [72]. Of note, ZnONPs have attractive antimicrobial activities that encourage their use as a therapeutic antimicrobial agent [73]. ZnO can induce bacterial death and/or inhibition through different mechanisms [74]. ZnO can directly interfere with bacterial cell integrity [75], induce the release of antimicrobial Zn^+2^ ions [76], and generate several reactive oxygen species such as hydrogen peroxide, hydroxyl, and superoxide free radicles, which penetrate the bacterial membranes, leading to bacterial damage and death [77,78,79]. Furthermore, ZnO nanoparticles may reduce bacterial attachment to biomedical surfaces [80]. Importantly, the combination of ZnO nanoparticles with other antibiotics can lead to enhanced antimicrobial properties, leading to a lower rate of bacterial resistance, shorter treatment durations, and lower drug concentrations used in treatment [81]. Venubabu Thati, et al. [82] noted that ZnO nanoparticles showed better antimicrobial properties when used in combination with other antibiotics such as cephalosporins and aminoglycosides.

### 3.9. Histological Analysis of the Skin Layers Treated with the Different Formulations

H&E-stained section examination of un-injured animals (negative control group) revealed that the skin consisted of the normal epidermis and the dermis. The epidermis was made up of four layers of stratified keratinized squamous epithelium: stratum basale, stratum spinosum, stratum granulosum, and stratum corneum [83]. Epidermal invaginations were interdigitated with dermal papillae to create the dermal–epidermal junction (DED). The dermis was made up of two layers: an exterior reticular layer rich in collagen, elastic fibers, and connective tissue cells, and an inner reticular layer rich in collagen, elastic fibers, and connective tissue cells (Figure 10a). In the untreated control group, the epidermis showed a significant reduction in thickness compared with group I, in addition to flattening of the dermal–epidermal junction (Figure 10b). Most epidermal cells showed vacuolated cytoplasm and darkly stained nuclei (Figure 10b). However, rats treated with AZM-ZnONPs showed an increased thickness of nucleated keratinocytes. Dermal papillae became deeper and more frequent (Figure 10c,d). Many keratinocytes appear with normal cytoplasm and were not vacuolated, whereas in AZM-ZnONP treatment, more improvement was noticed in the form of retaining the dermal thickness, and most of the cells appeared similar to those of the negative control group (Figure 10e).

With respect to the epidermal thickness, a significant decrease in the skin epidermal thickness was observed in the untreated group. However, rats that received the AZM-ZnONP formulation achieved the highest improvement in epidermal thickness compared to animals that were treated with either AZM or ZnO (Figure 10f).

Furthermore, negative control sections stained with Masson’s trichrome revealed copious thick collagen bundles in both the papillary and reticular dermis (Figure 11a). Fine collagen fibers with larger interspaces were seen in the papillary layer of the dermis in the untreated control group. Collagen bundles were also disorganized, and congested blood vessels were observed. In AZM- and ZnO-treated rats, there was a slight increase in the level of collagen fibers. However, in the AZM-ZnONP-treated group, the collagen fibers appeared in normal form and distribution (Figure 11a–e). Regarding the dermal measurements, the percentages of collagen fibers were significantly improved upon treatment. However, the highest levels of collagen fibers were achieved in the AZM-ZnONP-treated group (Figure 11f).

## 4. Conclusions

ZnONPs loaded with AZM provided a topical formulation that could markedly promote wound healing of infected wounds. Optimization of the formulation variables was beneficial in the selection of the most effective formula. The loading of AZM onto the surface of ZnONPs provided a healing effect that exceeded the healing effect of either drug alone. The observed in vitro and in vivo antibacterial and wound-healing activities for AZM-ZnONP-treated animals indicates that the developed formulation can offer a better alternative for the treatment of skin infections caused by resistant bacteria. Future studies will focus on testing the clinical efficacy of AZM-ZnONPs in the treatment of chronic diabetic wounds.

## Data Availability

Not applicable.

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
