# Peer review of "Tailoring of Novel Azithromycin-Loaded Zinc Oxide Nanoparticles for Wound Healing"

_pharmaceutics, 2022, doi:10.3390/pharmaceutics14010111_

Round 1

Reviewer 1 Report

Comments

Title: Tailoring of Novel Azithromycin-Loaded Zinc Oxid Nanoparticles for Wound Healing

The work presented in this article meets sufficient data of interest to Pharmaceutics. It represents a relatively study design of ZnO Nanoparticles loaded with azithromycin as an antibacterial compound for wound dressing application. The composite was well evaluated in vitro and in vivo.

I have some recommendations for the authors as below before acceptance:

  1. Minor English editing is required for improving readability.
  2. I recommend that authors please bring your chemical evaluations such as FTIR, XRD, and EDS in one figure as “chemical evaluations.”
  3. Would you please prepare a table for XRD and FTIR and explain the major peaks?
  4. In figure 1, what is X axisis and Y axisis? Please add suficent dada.
  5. For table 4, please prepare the relevant figure.
  6. In table 4 authors mentioned “Inhibition zone diameter (mm)” mm is unit of which part? All numbers have a unit as μg/ml. Please redesign the table.
  7. “ml” should change to “mL”

8. Would you please mix Figures 11 and 12 because the only difference is magnification? 

Author Response

Reviewer #1:

 Many thanks for worthy comments that enhance our manuscript quality, and we give detailed responses here:

1_Minor English editing is required for improving readability

Response: We apologize, the manuscript was revised thoroughly for typing mistakes.

2-I recommend that authors please bring your chemical evaluations such as FTIR, XRD, and EDS in one figure as “chemical evaluations.”

Response: We merged figures of FTIR, XRD and EDS in one figure (Figure 1A-C).

3- Would you please prepare a table for XRD and FTIR and explain the major peaks?

Response: We added two tables for XRD data in the supplementary data information (Table S1) file and FTIR in the main document (Table 2) and explained the major peaks.

4- In figure 1, what is X axisis and Y axisis? Please add suficent dada.

Response: X axis is 2θ theta angle of diffraction and y axis is intensity and we added to the figure 1A

5_For table 4, please prepare the relevant figure.

Response: We thank the reviewer for the suggestion. In response to reviewer comment, table 4 was reformulated and data were shown in figure 8. In addition, representative figures of the inhibition zones were added to the columns to make the results more clear.

6_In table 4 authors mentioned “Inhibition zone diameter (mm)” mm is unit of which part? All numbers have a unit as μg/ml. Please redesign the table.

Response: We apologize for this typing mistake. ug/ml was corrected to be mm. In addition table 4 was reformulated and data were shown as figure 8. Also, representative figures of the inhibition zones were added to the columns to make the results more clear.

7- “ml” should change to “mL”.

Response: All the manuscript had been checked and ml changed to mL and also the concentration units were corrected to be µg/mL instead of ug/mL.

  1. Would you please mix Figures 11 and 12 because the only difference is magnification? 

Response: Thank you. In response to comments, figures 11 and 12 were revised and magnified images were supplied as magnification sections in figure 10 (according to the new order of tables). Accordingly, figure 12 was removed.

Reviewer 2 Report

Although the paper is interesting, I recommend authors to improve the paper according to following lines:

1- A schematic image to represent the mechanism of adsorption of Azithromycin on to Zinc Oxide nanoparticle should be represented.

2- It would be great if authors provide the images of the plates used to culture bacteria and the inhibition zone for different samples.

3- Recent studies on synthesis of zinc based nanoparticle using other methods is recommended. Several studies are presented for this purpose rather than calcination at high temperature including biomineralization. In this regard, a study that can be considered at introduction:  Biomineralization‐Inspired Green Synthesis of Zinc Phosphate‐Based Nanosheets in Gelatin Hydrogel, International Journal of Applied Ceramic Technology 13 (6), 1069-1073 (2016).

4- The mechanism of antibacterial properties of Azithromycin on to Zinc Oxide nanoparticle should be discussed further.

Author Response

Reviewer #2:

Many thanks for worthy comments that enhance our manuscript quality, and we give detailed responses here:

1- A schematic image to represent the mechanism of adsorption of Azithromycin on to Zinc Oxide nanoparticle should be represented.

Response: Thank you for the suggestion. A schematic image that represents the mechanism of adsorption of Azithromycin onto Zinc Oxide nanoparticles was supplied.

2- It would be great if the authors provide the images of the plates used to culture bacteria and the inhibition zone for different samples.

Response: We thank the reviewer for the suggestion. In response to reviewer comment, table 4 was reformulated and data were shown as figure 8. In addition, representative figures of the inhibition zones were added to the columns which made the results more clear for readers.

3- Recent studies on synthesis of zinc based nanoparticle using other methods is recommended. Several studies are presented for this purpose rather than calcination at high temperature including biomineralization. In this regard, a study that can be considered at introduction:  Biomineralization‐Inspired Green Synthesis of Zinc Phosphate‐Based Nanosheets in Gelatin Hydrogel, International Journal of Applied Ceramic Technology 13 (6), 1069-1073 (2016).

Response: Different recent references for several synthesis approaches of ZnO-NPs were added according to your recommendations (references # 14-18). Thank you for guidance.

4- The mechanism of antibacterial properties of Azithromycin on to Zinc Oxide nanoparticle should be discussed further.

Response: Thank you. In reponse to reviewer comment, we elucidated the advantage of zinc oxide nanoparticles and its combination with azithromycin. The following section was revised and added to the manuscript

“Antibacterial activities of zinc oxide nanoparticles and AZM has received significant global interest particularly by the application of nanotechnology to develop particles in the nanometer range. According to our results, the combination of AZM and ZnONPs resulted in an improved antimicrobial and wound healing potencies, which may result from the enhanced physicochemical characteristics of the formulation. Analyzing the MIC of the different formulations showed that the AZM-ZnONPs formulation was very effective against both MRSA and E coli and the MIC was reduced markedly compared to each of them if used alone. Similarly, Aljihani, et al. [32] reported that Liposomal AZM and liposomal AZM/N-acetyl cysteine markley improved the MIC of AZM against E coli. Another study showed that AZM nanoparticles produced the same inhibitory effect against E coli at 8 times lower concentration compared to free AZM [53]. In another approach, cefixim and AZM were synthesized by anti-solvent precipitation and the synthesized nanoparticles showed better antibacterial activities against S. aureus, E. coli, Salmonella typhi and Shigella as tested by agar diffusion assays than the parental drugs [54]. Furthermore, PLGA-loaded AZM nanoparticles had improved antibacterial potential for treating intracellular infections compared to free drug [55]. Of note, ZnONPs have attractive antimicrobial activities which encourages its use as a therapeutic antimicrobial agent [56]. ZnO can induce bacterial death and/or inhibition through different mechanisms [57]. ZnO can directly interfere with bacterial cell integrity [58], induce the release of the antimicrobial Zn+2 ions [59], and generate several reactive oxygen species such as hydrogen peroxide, hydroxyl, and superoxides free radicles, which penetrates the bacterial membranes leading to bacterial  damage and death [60-62]. Furthermore, ZnO nanoparticles may reduce the bacterial attachment of to biomedical surfaces [63]. Importantly, the combination of ZnO nanoparticles with other antibiotics can lead to enhanced antimicrobial properties leading to a lower rate of bacterial resistance, shorter treatment durations and lower drug concentrations used in treatment [64]. Venubabu Thati, et al. [65] noted that ZnO nanoparticles showed better antimicrobial properties when used in combination with other antibiotics such as cephalosporins and aminoglycosides.”

Round 2

Reviewer 2 Report

The paper is acceptable now.